# Synbiotics effects of d-tagatose and *Lactobacillus rhamnosus GG* on the inflammation and oxidative stress reaction of *Gallus gallus* based on the genus of cecal bacteria and their metabolites

**Yuanqiang Lv**[©], **Jie Chu**[*©], **Xiaoxiao Zhang**[‡], **Xuan Li**[‡], **Aijiao Yin**[‡], **on behalf of The Industrial Microbiology Laboratory**[¶]

Qilu University of Technology (Shandong Academy of Sciences), Jinan, Shandong, China

© These authors contributed equally to this work.
‡ XZ, XL and AY also contributed equally to this work.
¶ Membership of the Industrial Microbiology Laboratory is provided in the Acknowledgments.
* chujie6532@163.com

**Data Availability Statement:** All data of this study in China National Center for Bioinformation

## Abstract

### Backgrounds

Abuse of feed supplement can cause oxidative stress and inflammatory responses in *Gallus gallus*. Synbiotics are composed of prebiotics and probiotics and it possess huge application potentials in the treatment of animal diseases.

### Methods

This study examined the effect of d-tagatose on the probiotic properties of *L. rhamnosus GG*, *L. paracasei*, *and S. lactis* so as to screen the best synbiotic combinations. Treat *Gallus gallus* exhibiting oxidative stress and immune response caused by aflatoxin b1 with optimal synbiotics for 14 days, detect the changes of inflammatory markers and oxidative stress markers of *Gallus gallus* using qRT-PCR, and identified the intestinal bacteria genera and their metabolites in the cecum of *Gallus gallus* using gut microbiota and metabolomics analysis.

### Results and conclusion

The results indicated that oxidative stress and immune response factor expressions quantity in Gallus gallus decreased significantly after 14 days of treatment, compared with model group, the low-dose treatment group's SOD1, SOD3, GPX1, GPX2, GSR, H6DP, and HO-1 genes in liver were downregulated by 36.03%, 40.01%, 45.86%, 40.79%, 37.68%, 25.04%, and 29.89%, the IL-1, IL-2, IL-4, IL-6, IgA, IgM, and IgG genes in blood and spleen were downregulated by 26.59%, 34.19%, 21.19%, 28.18%, 35.93%, 12.67%, 21.81 and 35.93%, 22.85%, 21.19%, 28.78%, 35.93%, 15.36%, 29.73%. The intestinal bacteria genera and metabolomics analysis results indicated that the abundance of beneficial bacteria genus

(CNCB), and the link is: https://ngdc.cncb.ac.cn/omix/preview/GBnactJE.

**Funding:** The author(s) received no specific funding for this work.

**Competing interests:** The authors have declared that no competing interests exist.

was up-regulated, and the proportion of pathogenic bacteria genera decreased. The amount of beneficial metabolites associated with antioxidant and anti-inflammatory effects was upregulated. The synbiotic composed of d-tagatose and L. rhamnosus GG can treat oxidative stress and immune response by altering the structure of intestinal bacteria genera and the production of metabolites.

## Introduction

Feed additives and high dietary energy intake increase the growth rate of *Gallus gallus* and reduce feeding costs [1]. However, it also significantly affects the intestinal flora [2], inducing oxidative stress and an inflammatory response [3]. Since China banned antibiotics in animal feed, the chicken mortality rate has increased from 2.9% to 4.2% [4], challenging the poultry industry. Intestinal flora helps maintain intestinal homeostasis [5]. *Lactobacillus* exhibits excellent gut colonization ability [1, 6] and bile and gastric acid tolerance [7]. They also inhibit the growth of pathogenic bacteria [8, 9] by secreting short-chain fatty acids [10, 11] to reduce intestinal pH. More importantly, *Lactobacillus* able to produce antioxidant enzymes, including catalase (CAT), superoxide dismutase (SOD), and glutathione peroxidase (GSH) [12]. Moreover, *Lactobacillus* causes the host to produce antioxidant enzymes. Studies show that enolase A1 produced by *L. plantarum* is involved in an immunoreaction [13].

Probiotics, prebiotics and synbiotics have a wide range of applications of treating and improving animal health [14, 15], and Table 1 lists the definition and role of them. d-Tagatose is a low-calorie rare sugar produced by the hydrolysis, small intestine absorbs and utilizes only 20% of d-tagatose based on its unique structure, and unabsorbed d-tagatose tilized by microbial colonizing in the colon [16–19]. Moreover, d-tagatose promotes probiotic colonization in the gut, such as *Lactobacillus*, and inhibits the growth of pathogenic bacteria by attenuating glycolysis and its downstream metabolism [20], Thus, d-tagatose significantly enhances intestinal flora diversity [18]. Several studies have been conducted on d-tagatose's in vitro probiotic properties over the past few years. d-Tagatose can promote the growth and intestinal colonization of *L. rhamnosus GG* by activating the phosphotransferase system-mediated tagatose metabolism network [17]. However, d-tagatose's in vivo probiotic properties still need to be verified by animal experiments.

This study takes d-tagatose as a prebiotic and combined it with probiotics (*S. lactis*, *L. paracasei*, *L. rhamnosus GG*) to investigate the in vitro prebiotic characteristics such as growth characteristics, antioxidant capacity, biofilm formation ability, and inhibitory effect on pathogenic bacteria growth of d-tagatose. Then screened for optimal synbiotic of d-tagatose and *Lactobacillus* based on the above experimental results. Thereafter, this study examined the ameliorating effect of synbiotics on the oxidative stress and immune response of *Gallus gallus*

**Table 1. Introduction of probiotics, prebiotics and synbiotics.**

|  | **Probiotics [14, 15]** | **Prebiotics [15]** | **Synbiotics [15]** |
|---|---|---|---|
| Definition | Live microorganisms that are beneficial to the host and improve the of gut microbiota. | Edible substances that are difficult to be absorbed by host and beneficial to host. | Mixed preparations of live probiotics and prebiotics. |
| Role | Regulating the gut microbiota and promoting host health by colonizing in intestine. | Selectively stimulating growth of beneficial bacteria in the intestine, thus having a beneficial impact on host. | Enhancing host health by regulating gut microbiota and promoting beneficial active substances' production of probiotics. |

and its impact on gut microbiota and their metabolic products based on RT-qPCR, cecum microbiota genera and metabolomics analysis.

## Materials and methods

### Abbreviation

All abbreviations used in the study were listed in Table 2.

### Materials

The Green Land Farming Professional Cooperative (Suqian, China) provided seven-day-old *Gallus gallus* (NCBI reference sequence: GCF_016699485.2.), and the breed is *Sanhuang chicken*. The standard feed was purchased from Guangyuan Feed Co., Ltd (Jinan, China). *L. rhamnosus GG*, *L. paracasei*, *S. lactis*, *E. coli*, *C. albicans*, and *S. aureus* were obtained from the Institute of Biology, Shandong Academy of Sciences. d-Tagatose was purchased from Shanghai Aladdin Biochemical Technology Co., Ltd (Shanghai, China). The free radical of 2,2-diphenyl-1-picrylhydrazyl (DPPH), 2,2'-azinobis-3-ethylbenzothiazoline-6-sulfonic acid (ABTS), hydroxyl (·OH) were provided by Jinan Fannuo Chemical Co., Ltd (Jinan, China). Luria-Bertani (LB) medium and de Man, Rogosa and Sharpe (MRS) medium were provided by Qingdao Haibo Biotechnology Co., Ltd (Qingdao, China).

### Effect of d-tagatose on the growth and lactate production of *Lactobacillus*

*L. rhamnosus GG*, *L. paracasei*, and *S. lactis* were grown in MRS-Tag and MRS-Blank at 37°C for 16 h. Regular sampling and $OD_{600}$ measurements were performed. Lactic acid concentration was determined using a biosensor (Institute of Biology, Shandong Academy of Sciences, Jinan, China). The addition amount of d-tagatose in MRS-Tag is 2% according to the content of glucose in MRS medium.

**Table 2. Full name and abbreviation.**

| Full name | Abbreviation |
|---|---|
| *Lactobacillus rhamnosus* | *L. rhamnosus* |
| *Lactobacillus paracasei* | *L. paracasei* |
| *Streptococcus.lactis* | *S. lactis* |
| *Escherichia coli* | *E. coli* |
| *Candida albicans* | *C. albicans* |
| *Staphylococcus aureus* | *S. aureus* |
| *Lactobacillus plantarum* | *L. plantarum* |
| Luria Bertani medium | LB medium |
| Aflatoxin B1 | AFB1 |
| De Man Rogosa Sharpe medium | MRS medium |
| De Man Rogosa Sharpe medium contain 2% d-tagatose | MRS-Tag |
| De Man Rogosa Sharpe medium without any sugar | MRS-Blank |
| Diphenylpicrylhydrazine | DPPH |
| Hydroxyl Radical | ·OH |
| 2,2'-Azinobis-(3-ethylbenzothiazoline-6-sulfonic acid) | ABTS |
| Weight gain rate | WGR |
| Specific growth rate | SGR |
| Feed efficiency rate | FER |

### Effects of d-tagatose on the antibacterial activity of *Lactobacillus*

*E. coli*, *C. albicans*, *and S. aureus* were grown in LB broth at 37˚C, 180 rpm for 16 h, and 100 μL above fermentation broth was inoculated onto MRS-Tag agar. Filter paper (R = 0.5cm) was placed in the center of the MRS-Tag agar, and 20 μL fermentation broth of *L. rhamnosus GG*, *L. paracasei*, or *S. lactis* was added to filter paper. The MRS-Tag were incubated at 37˚C for 24 h, and the inhibition zone was assessed.

### Effects of d-tagatose on the antioxidant activity of *Lactobacillus*

DPPH clearance was determined according to the method described by Najmeh et al. [21]. ABTS radical scavenging ability was measured according to the method described by Bożena et al. [22]. The method described by Zheng et al. measured hydroxyl radical clearance [23]. The cultivates of *L. rhamnosus GG*, *L. paracasei*, *and S. lactis* were the same as described in effect of d-tagatose on the growth and lactate production of *Lactobacillus*.

### Effects of d-tagatose on biofilm formation in different *Lactobacillus*

The cultures of different *Lactobacillus* were diluted to $OD_{600} = 0.1$ with MRS-Tag or MRS-Blank, and 100 μl fermentation broth were pipetted into a 96-well plate and cultured at 37˚C for 16 h. Next, 200 μL 0.9% NaCl was used to wash the biofilm twice, and 250 μL 0.5% crystal violet-staining solution was added for 30 min. The plates were washed twice with 0.9% NaCl, crystal violet was added, the biofilms were dissolved by adding 100 μL 96% $C_2H_5OH$ [24], and the $OD_{590}$ was measured. The cultivates of *L. rhamnosus GG*, *L. paracasei*, *and S. lactis* were the same as those described in effect of d-tagatose on the growth and lactate production of *Lactobacillus*.

### Ethics statement

Animal feeding, handling, and sampling were carried out in accordance with the guidelines of the Laboratory Animal Ethics Committee of Qilu University of Technology (Shandong Academy of Sciences), try to minimize the pain of animals during the sampling process. The approval number for the animal experiment is: SWS20241111.

### Design of the animal experiments

As shown in Fig 1, a total of 40 seven-day-old *Gallus gallus* were adaptively fed for four days and randomly assigned to five groups. An oxidative stress and immune response model was built using feed supplemented with 50 μg/kg of AFB1 for three days for the treatment and model groups [25, 26]. The blank group was synchronously fed for three days with the feed. Subsequently, synbiotics composed of *L. rhamnosus GG* and d-tagatose was added in water for drinking. This study added *Lactobacillus rhamnosus GG* in water so that *Gallus gallus* could ingest *lactobacillus* followed the method of Shehata et al. with slight modifications [27], and Table 3 lists the grouping strategy of design of the animal experiments. *Gallus gallus* were fed in cartons indoors of the Institute of Biology, Shandong Academy of Sciences (Jinan, China), feeding environment was clean and maintained at an appropriate temperature, humidity, and light. *Gallus gallus* ate and drank ad libitum and the treatment period was from April 7, 2024 to April 20, 2024 lasting for 14 days. Period of sample collection was April 21, 2024, *Gallus gallus* were then fasted for 12 hours before sample collection.

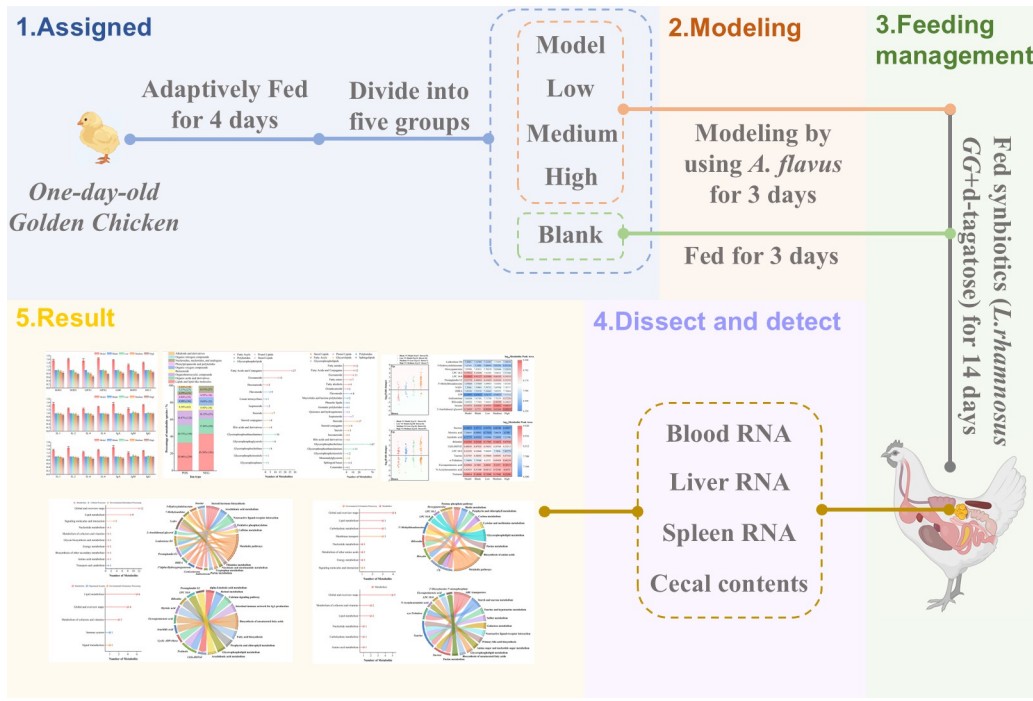

**Fig 1. The animal experimental protocol.**

## Effect of synbiotics on the growth of *Gallus gallus*

The weight of *Gallus gallus* in each group was measured to calculate growth performance and feed utilization based on the following formulae:

$$\text{Weight gain rate (WGR)}/\% = (\text{final weight} - \text{initial weight})/\text{initial weight} \times 100\%$$

$$\text{Specifc growth rate (SGR)}/\% = (\text{final weight} - \text{initial weight})/\text{duration} \times 100\%$$

$$\text{Feed efficiency rate (FER)} = (\text{final weight} - \text{initial weight})/\text{feed intake}$$

## Real-time quantitative PCR (RT-PCR)

Liver and spleen RNA were extracted using the Vazyme FreeZol Reagent (Vazyme Biotech Co., Ltd, Nanjing, China). Blood RNA was extracted using the Total RNA Extraction Kit (Solarbio Biotech Co., Ltd, Beijing, China). RNA purity and concentration were assessed first, and then the integrity was verified by gel electrophoresis. Subsequently, RNA was reverse-transcribed using the Vazyme Hiscript Q RT SuperMix for qPCR+gDNA wiper (Nanjing Vazyme Biotechnology Co, LTD, Nanjing, China). RT-PCR was carried out using the Vazyme ChamQ Universal SYBR qPCR Master Mix (Nanjing Vazyme Biotechnology Co, LTD). The gene

**Table 3. The grouping strategy of design of the animal experiments.**

|  | Blank group | Model group | Low-dose treatment group | Medium-dose treatment group | High-dose treatment group |
|---|---|---|---|---|---|
| d-tagatose | 0 | 0 | 0.5% | 1.0% | 1.5% |
| *L. rhamnosus GG* | 0 | 0 | $1.0 \times 10^5$ cfu/mL | $1.0 \times 10^5$ cfu/mL | $1.0 \times 10^5$ cfu/mL |

expression levels of inflammatory cytokines and oxidative stress cytokines were determined. S1 Table lists the fluorescent probe sequences used for RT-PCR.

## Analysis of the gut bacterial genus

Microbial DNA cecal content was extracted using a Starvio DNA stool Kit (Solarbio Biotech Co., Ltd, Beijing, China), Thereafter, DNA quality was assessed. The 16s DNA V4 region (16SV4 DNA) was cloned using the $F_{515}$ primer (5'-GTGCCAGCMGCCGCGGTAA-3') and $R_{806}$ primer (5'-GGACTACHVGGGTWTCTAAT-3'). The PCR products were purified by magnetic bead purification and sequenced on the NovaSeq6000 PE250 platform. After removing the barcoed sequences and primer sequences from the sample data, the sample reads were concatenated using flash to obtain the raw tags (Version 1.2.11, http://ccb.jhu.edu/software/FLASH/) [28]. The raw tags were filtered using fast (Version 0.23.1) for clean tag acquisition [29]. The abundance of the bacterial genera was analyzed using the pheatmap function. The relative abundance of *Bifidobacterium*, *Lactobacillus*, and *E. coli* in the cecal samples was determined according to the method of Gu et al. [30]. S2 Table lists the fluorescent probe sequences used for RT-PCR.

## Metabolomics analysis of the cecal contents

Cecal samples (200 mg) were placed into an EP tube, 200 μL of 80% $CH_3OH$ was added, and the mixture was stirred thoroughly. The samples were then centrifuged at 15000 rpm for 20 min at 4°C, and the supernatant was diluted to a final $CH_3OH$ content of 53%. The concentration of the various metabolites of the cecal samples was measured by ultra-high-performance liquid chromatography coupled with high-resolution mass spectrometry (UHPLC-MS/MS). The liquid chromatography conditions were as follows: chromatographic column: Hypersil Goldcolumn (100 × 2.1 mm, 1.9 um), flow rate: 0.2 mL/min, column temperature: 40°C, mobile phase A: 0.1% $CH_3COOH$, and mobile phase B: $CH_3OH$. Metabolite analysis used a 12-min linear gradient elution as follows: 0–1.5 min (A: B = 98:2), 1.5–3 min (A: B = 15:85), 3–10 min (A: B = 0: 100), and 10–12 min (A: B = 98:2). The mass spectrum conditions included the following: spray voltage: 3.5 kV, capillary temperature: 320°C, sheath gas flow rate: 35 psi, aux gas flow rate: 10 L/min, S-lens RF level: 60, and aux gas heater temperature: 350°C. Accurate qualitative and relative quantitative results of the metabolites were determined using the statistical software R (R version R-3.4.3), Python (Python 2.7.6 version), and CentOS (CentOS release 6.6).

## Results

### Effect of d-tagatose on the probiotic properties of *Lactobacillus*

*Lactobacillus* was cultured in MRS-Tag and MRS-Blank at 37°C for 16 h, the bacterial density ($OD_{600}$), lactic acid yield of fermentation broth was measured. As shown in Fig 2A, d-tagatose promoted the growth of *S. lactis*, *L. rhamnosus GG*, and *L. paracasei*, and d-tagatose also increased the production of lactic acid. As shown in Fig 2B, *lactobacillus* and pathogenic bacteria were cultured in MRS-Tag and MRS-Blank at 37°C for 16 h, a transparent zone appeared near the *Lactobacillus* bacterial colony, d-tagatose improved the antibacterial ability of *S. lactis*, *L. rhamnosus GG*, and *L. paracasei.*, and *L. rhamnosus GG* and *L. paracasei* did not inhibit the growth of *C. albicans*, whereas *C. albicans* grows weakly on MRS-Tag. As shown in Fig 2C, by detecting the free radical scavenging rate of the *L. lactis* fermentation supernatant and cell-breaking supernatant, this study found that d-tagatose enhanced the free radical scavenging ability of *S. lactis*, *L. rhamnosus GG*, and *L. paracasei*, including DPPH, ABTS, and ·OH.

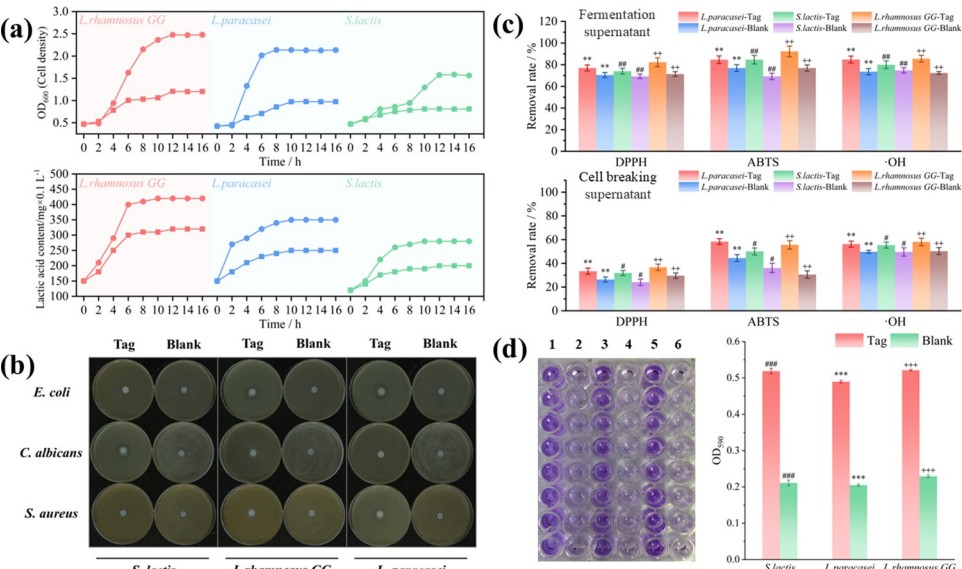

**Fig 2. Effect of d-tagatose on *Lactobacillus*.** The results of Fig 2C and D are expressed as mean values and standard errors of 3 biological replicates. Fig 2d (left), 1: *L. paracasei* grown in MRS-Tag, 2: *L. paracasei* grown in MRS-Blank, 3: *S. lactis* grown in MRS-Tag, 4: *S. lactis* grown in MRS-Blank, 5: *L.rhamnosus GG* grown in MRS-Tag, 6: *L.rhamnosus GG* grown in MRS-Blank. \*\*, # # and + + is P < 0.01, \*\*\*, # # # and + + + is P < 0.005.

Fig 2D shows that d-tagatose promotes the biofilm formation of *S. lactis*, *L. rhamnosus GG*, and *L. paracasei*. The experimental results of the free radical scavenging ability and biofilm formation are all presented in the form of mean ± standard error of three biological replicates.

## Synbiotics improve the growth of *Gallus gallus*

*Gallus gallus*'s weight and food intake were assessed during treatment. The WGR, SGR, and FER were calculated based on the initial weight / kg, final weight / kg, and food intake / kg (Fig 3). Synbiotics improved *Gallus gallus*'s growth and feed utilization when treated for 14 days. The treatment group's final weight ($m_i$) was significantly higher than the blank and model groups. The final weight ($m_i$), WGR, SGR, and FER of *Gallus gallus* improved with increased d-tagatose content in the synbiotics (P < 0.1). All statistical tests and their significance level analyses in animal experiment were completed using GraphPad Prism 9.

## Synbiotics relieve the oxidative stress and inflammatory response of *Gallus gallus*

It has been reported that AFB1 can increase the release of inflammatory factors such as interleukin-1β (IL-1β) and interleukin-18 (IL-18) etc. in microglia and promote the occurrence of apoptosis [25], and it also can significantly upregulate a variety of inflammatory factors in porcine intestinal cells, including interleukin (IL) series, tumor necrosis factor-α (TNF-α) and oxidative stress factors [26]. Fig 4A shows the changes in the liver's relative expression of oxidative stress cytokines. The expression of the $SOD_1$, $SOD_3$, $GPX_1$, $GPX_2$, $GSR$, $H6DP$, and $HO$-$1$ genes associated with oxidative stress increased significantly after *Gallus gallus* oxidative stress model setting-up (P < 0.001). Fig 4B and 4C show the relative expression changes of inflammatory cytokines in the spleen and blood. The expression of the *IL-1*, *IL-2*, *IL-4*, *IL-6*, *IgA*, *IgM*, and *IgG* genes related to the inflammatory response was upregulated significantly after *Gallus gallus* inflammatory model setting-up (P < 0.001). Moreover, oxidative stress

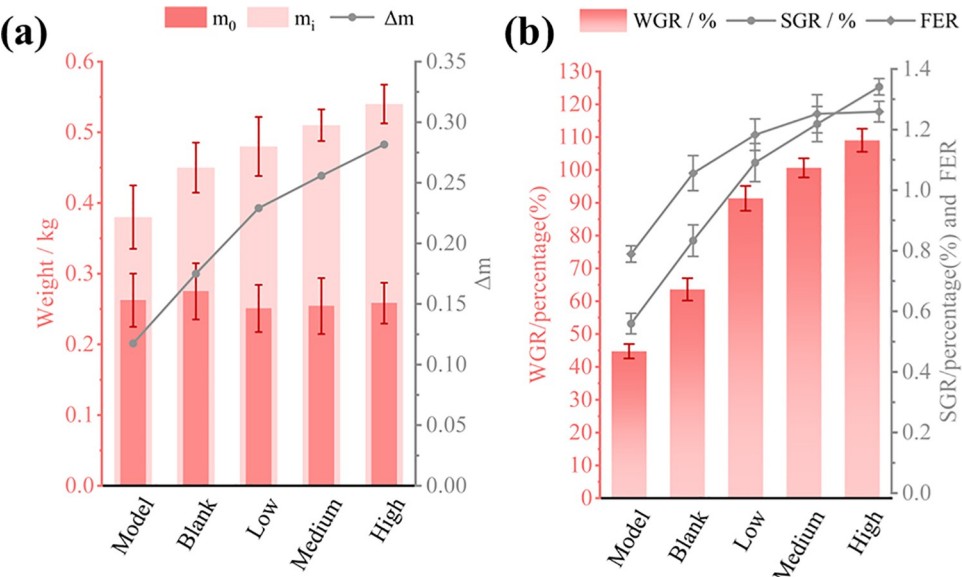

**Fig 3. Synbiotics improve the growth of *Gallus Gallus*.** Fig 3A is the changes in body weight of *Gallus gallus*, $m_0$: initial body weight / kg, $m_i$: final body weight / kg, $\Delta m$: final body weight—initial body weight / kg. Fig 3B is the WGR, SGR, and FER of *Gallus gallus*, WGR is the weight gain rate / % of *Gallus gallus*, SGR is the specific growth rate / %, and FER is the feed efficiency rate. The results in Fig 3 are expressed as mean values and standard errors of five *Gallus gallus* per group.

cytokines and inflammatory cytokine expression decreased significantly after feeding for 14 days using synbiotics. Increased d-tagatose content in the synbiotics enhanced the treatment effect.

## Synbiotics attenuate the cecal microbiota of *Gallus gallus*

Subsequent to the PCR amplification of the cecal microbial DNA with primers specific to the 16S DNA V4 region, sequencing was carried out using the NovaSeq6000 PE250 platform. The reads of each sample were assembled using FLASH (Version 1.2.11, available at http://ccb.jhu.edu/software/FLASH/). Subsequently, the fastp software (Version 0.23.1) was utilized to perform a rigorous filtering process on the raw tags derived from the assembly, thereby generating high-quality tags data. Use the DADA2 module in the QIIME2 software (Version QIIME2-202202) to perform noise reduction on Clean Tags, so as to obtain the final ASVs (Amplicon Sequence Variants) and feature tables. Subsequently, use the micro_NT database in the QIIME2 software to carry out species annotation. The annotation results were sorted according to abundance at different classification levels. A stacked percentage column chart was generated using the top eight bacteria genera with the largest abundance in each group at the genus classification level. As shown in Fig 5A, the bacterial genera with the highest relative abundance at the bacterial genus level were *Bacteroides*, *Ligilactobacillus*, *Phascolarctobacterium*, *and Prevotellaceae*. The abundance of *Ligilactobacillus* and *Bacteroides* was significantly higher than the model group after treatment (P < 0.05), and *Prevotellaceae* and *Desulfovibrio*'s abundance was lower than the model group. Notably, with the increase of d-tagatose content in the synbiotics, the abundance of the bacterial genera *Phascolarctobacteria*, *Ligilactobacilli*, and *Bacteroides* gradually increased. In contrast, the abundance of *Prevotelaceae* and *Desulfovibrio* decreased. This study also detected the cecal flora of *Gallus gallus* using RT-PCR to examine the changes in *Bifidobacterium*, *Lactobacillus*, and *E. coli* abundance. Fig 5B shows that the

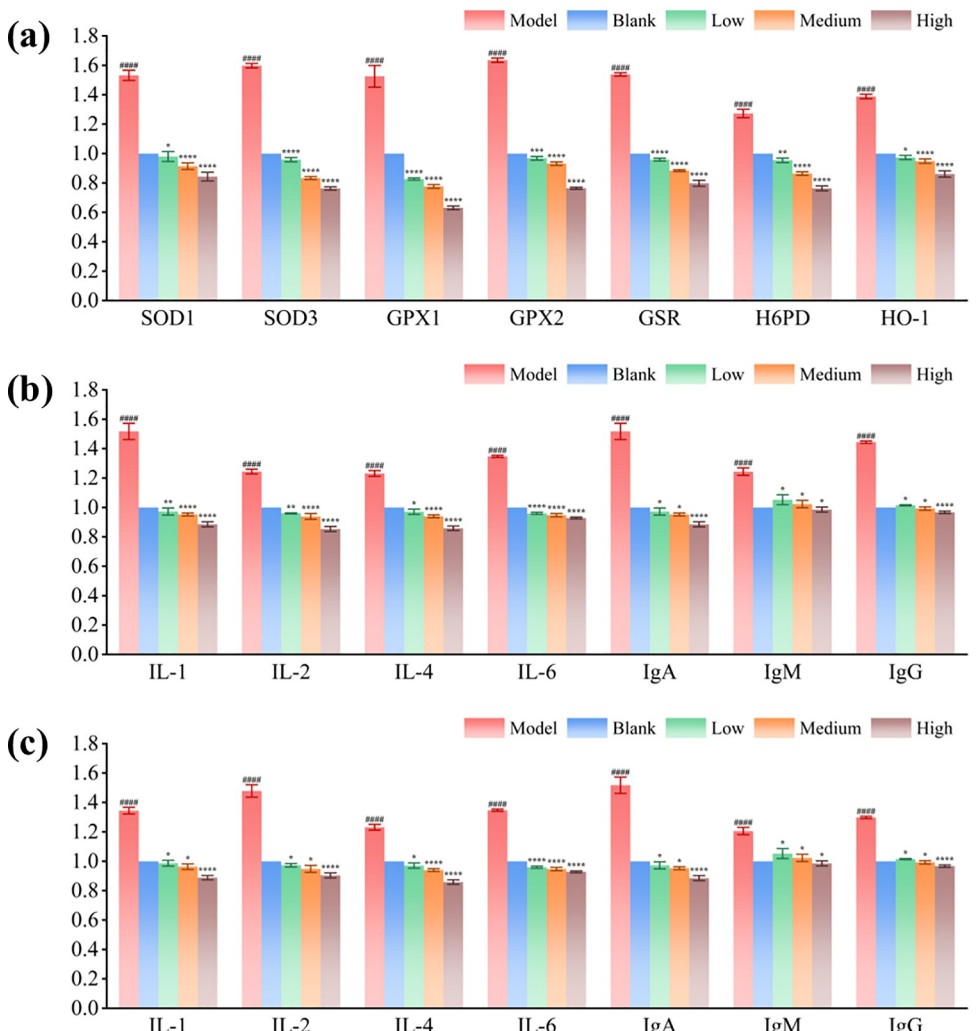

**Fig 4. Expression of oxidative stress cytokine and inflammatory cytokine genes.** *IL-1*, *IL-2*, *IL-4*, *IL-6*: interleukins genes. *IgA*, *IgM IgG*: immunoglobulin genes. The model and treatment groups are normalized by blank group. The results in Fig 4 are expressed as mean values and standard errors for five animals per group. * P < 0.05, ** P < 0.01, *** P < 0.005, **** P < 0.001.

abundance of beneficial bacteria, such as *Bifidobacterium* and *Lactobacillus*, increased after treatment (P < 0.05). The abundance of *E. coli* was decreased (P < 0.05).

## Metabolomics analysis of *Gallus gallus* cecal content

UHPLC-MS/MS analysis was performed on the fermentation broth supernatant with the utilization of QExactive™HF/QExactive™HF-X (ThermoFisher, Germany). The qualitative and quantitative analysis of metabolites was realized by employing Python-3.5.0 software, classification of metabolites was attained through the utilization of the R package (R-3.4.3), differential analysis was conducted by leveraging Python-3.5.0 software in combination with the R package (R-3.4.3) and KEGG enrichment analysis was achieved by means of Python-3.5.0 software and the R package (R-4.0.3). This study identified 1926 metabolites from *Gallus gallus* cecal samples, including 1408 positive ion metabolites and 518 negative ion metabolites, and selected 721 significantly altered positive and 323 negative ion metabolites after filtering the metabolites based on an FC <0.833 and P-value <0.05. As shown in Fig 6A, lipids and lipid-

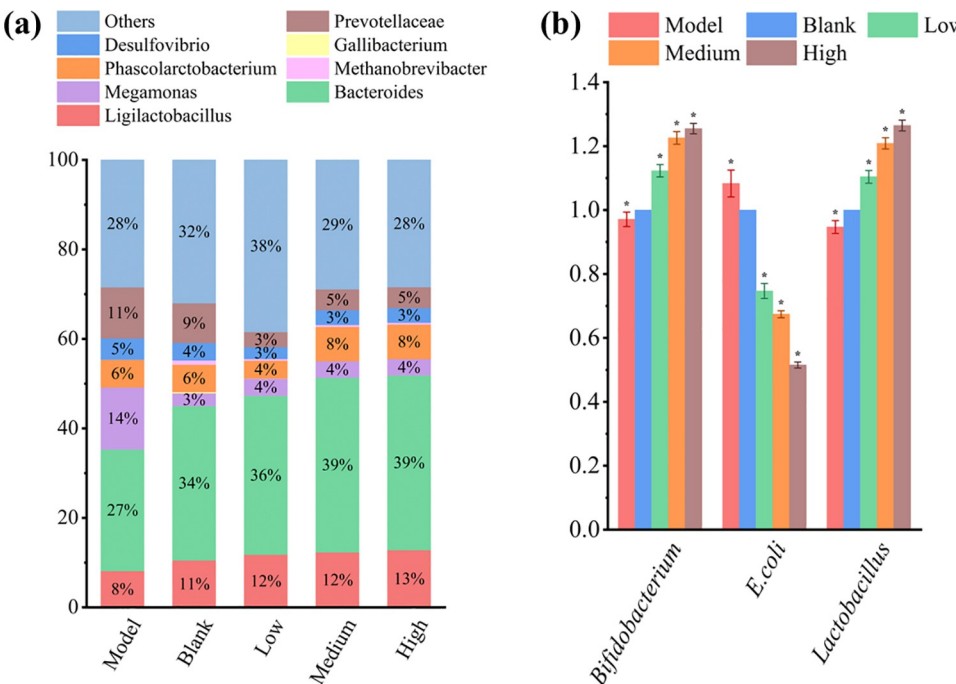

**Fig 5. The effect of synbiotics on the cecum bacterial genus of *Gallus Gallus*.** The concentration of all DNA templates used for RT-PCR was 20 ng/uL. The blank group normalized the model and treated groups, and the results were expressed as mean values and standard errors for five animals per group. *: P < 0.05.

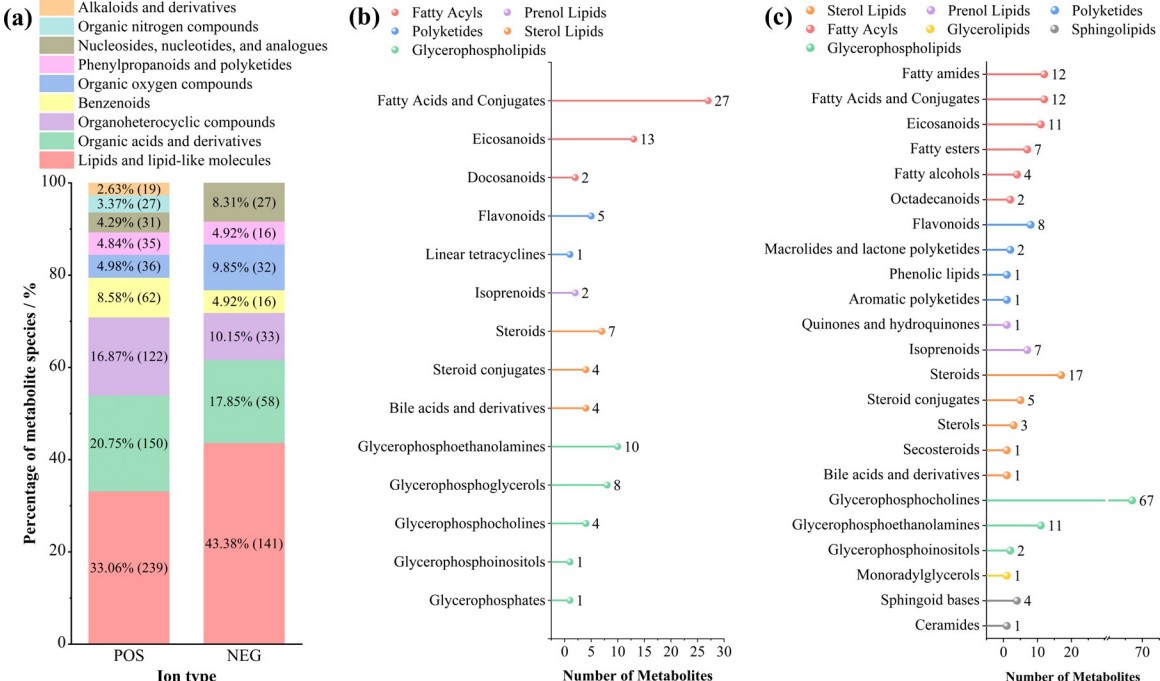

**Fig 6. Metabolite clustering analysis of the cecal content.** Fig 6B, 6C shows the lipid two-level classification chart of the negative ion and positive ion metabolites.

like molecules were the most abundant, accounting for 33.06% (239) of the positive ion metabolites and 43.08% (141) of the negative ion metabolites. Organic acids and their derivatives accounted for 20.75% (150) of the positive ion metabolites and 17.85% (58) of the negative ion metabolites. The metabolites associated with organoheterocyclic compounds accounted for 16.87% (122) of the positive ion metabolites and 10.15% (33) of the negative ion metabolites. We annotated the lipid metabolites using LIPID MAPS, and they were divided into the following eight categories: fatty acyls, glycerolipid, glycerophospholipid, sphingolipids, sterol lipids, prenol lipids, saccharolipids, and polyketides. Fig 6B and 6C show that we found 67 glycerophosphocholines, 17 steroids, 12 fatty acids, and conjugates in the positive ion metabolites. The negative ion metabolites contained 27 fatty acids and conjugates and 13 eicosanoids.

As shown in Fig 7A, there were 34 significantly different negative ion metabolites in the blank group compared with the model group based on a metabolomics analysis, which consisted of 9 upregulated and 25 down-regulated metabolites. This study also identified 19 significantly different negative ion metabolites in the medium-dose treatment group compared with the low-dose treatment group, which consisted of 12 upregulated and 7 down-regulated metabolites. Fig 7B shows 68 significantly different positive ion metabolites, including 33 upregulated and 35 down-regulated metabolites in the blank group compared with the model group. Similarly, there were 48 significantly different positive ion metabolites, encompassing 22 upregulated and 26 down-regulated metabolites. Based on Fig 7C, the differences in the negative ion metabolites were evident in the different groups. The upregulated negative ion metabolites included sucrose, arachidic acid, taurine, tretinoin, N-acetylneuraminic acid,

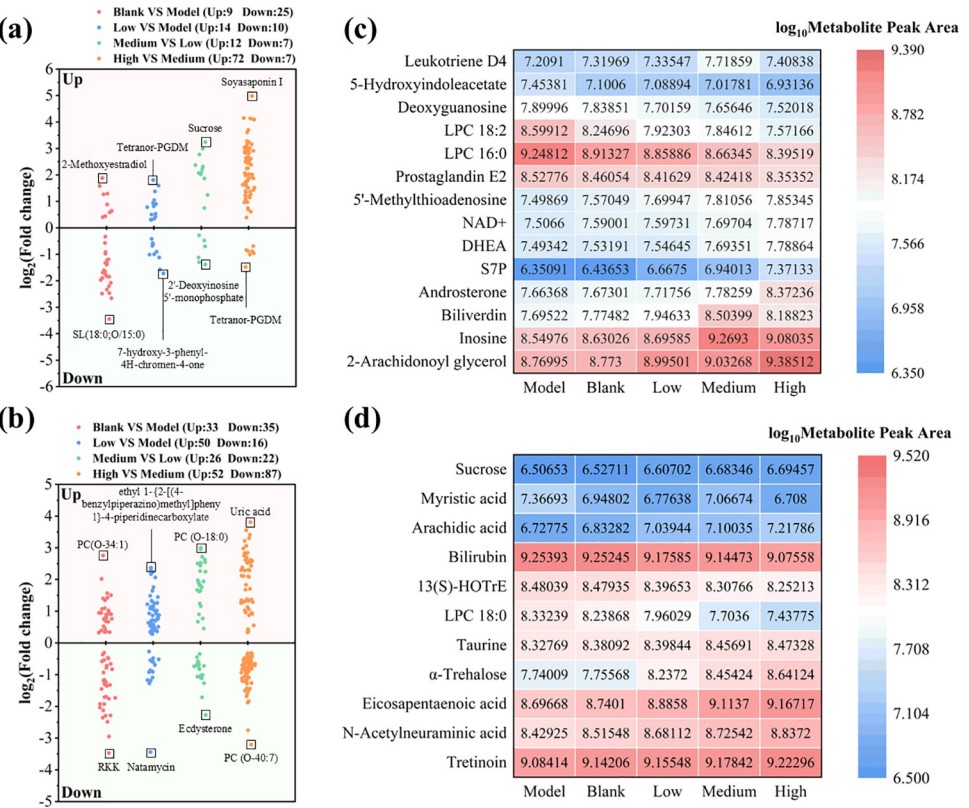

**Fig 7. The differences in metabolites.** (a) The volcano plot of significantly different negative ion metabolites is shown. (b) The volcano plot of significantly different positive ion metabolites. (c) Heat map of significantly different negative ion metabolites. (d) Heat map of significantly different significantly different positive ion metabolites.

eicosapentaenoic acid, α-trehalose. The down-regulated negative ion metabolites included myristic acid, lysophosphatidyl choline 18:0 (LPC 18:0), 13(S)-HOTrE, and bilirubin. The upregulated positive ion metabolites included leukotriene D4, 5'-methylthioadenosine, nicotinamide adenine dinucleotide (NAD$^+$), dehydroepiandrosterone (DHEA), sedoheptulose 7-phosphate (S7P), androsterone, 2-arachidonoyl glycerol, inosine, and biliverdin. The down-regulated positive ion metabolites included lysophosphatidyl choline 16:0 (LPC 16:0), lysophosphatidyl choline 18:2 (LPC 18:2), deoxyguanosine, 5-hydroxyindoleacetate, and prostaglandin E2 (Fig 7D).

## Metabolites KEGG enrichment analysis of *Gallus gallus*'s cecal content

KEGG enrichment analysis results for positive ion metabolites show that the metabolites were primarily related to "Lipid metabolism," "Signaling molecules and interaction," "Carbohydrate metabolism," and "Membrane transport" etc. (Fig 8). Moreover, KEGG enrichment analysis revealed that the negative ion metabolites were primarily associated with "Lipid metabolism" and "Metabolism of cofactors and vitamins" etc. (Fig 9).

## Discussion

In this study, d-tagatose significantly promoted the growth of *S. lactis, L. rhamnosus GG, and L. paracasei. L. rhamnosus* is derived from the human gut and has been used as a probiotic [31], and d-Tagatose is generally recognized as safe [32]. d-Tagatose showed a superior growth-promoting effect on *L. rhamnosus GG*, this is consistent with the reported that d-tagatose can enhance *L. rhamnosus GG* [17]. The unexcepted ability to effectively utilize d-tagatose of *Lactobacillus* prompted us to study the effect of d-tagatose on their in vitro probiotic

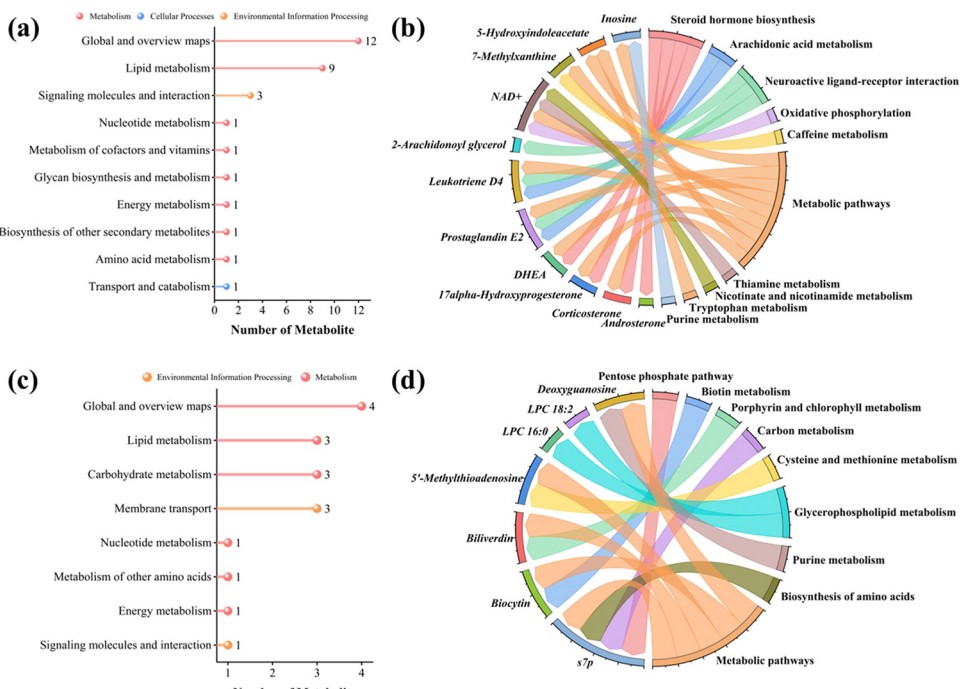

**Fig 8. KEGG enrichment analysis of positive ion metabolites.** Fig a and b are the KEGG enrichment analysis of positive ion metabolites between blank and model group. Fig c and d are the KEGG enrichment analysis of positive ion metabolites between medium and low-dose treatment group.

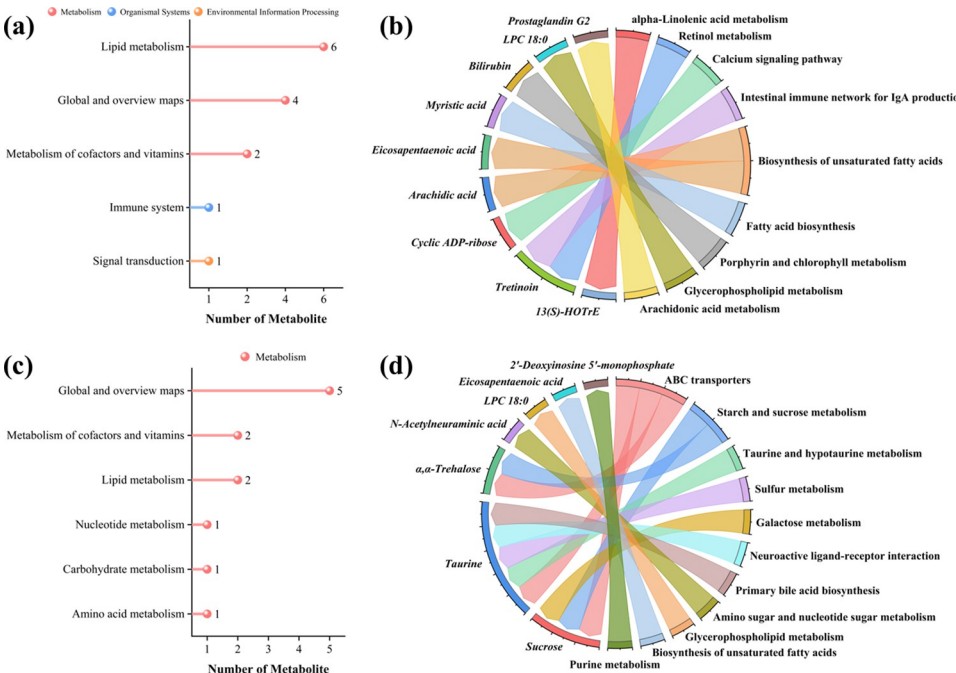

**Fig 9. KEGG enrichment analysis of negative ion metabolites.** Fig a and b are the KEGG enrichment analysis of negative ion metabolites between blank and model group. Fig c and d are the KEGG enrichment analysis of negative ion metabolites between medium and low-dose treatment group.

properties. *L. rhamnosus GG* had the greatest ability to produce lactic acid, antibacterial substances, antioxidants, and biofilm. Many metabolites that *Lactobacillus* generates exhibit bacteriostasis, including antimicrobial peptides [33], phenylacetic acid [34], lactic acid, acetic acid [35], and citric acid. In the present study, this study only measured lactic acid production by *S. lactis*, *L. rhamnosus GG*, and *L. paracasei* and the effect of d-tagatose. Further studies are needed to determine whether they can produce other antibacterial substances. Biofilm is often considered an essential indicator of intestinal bacterial colonization [36]. Intestinal mucosal colonization of *Lactobacillus* is a prerequisite for producing an effective and sustainable beneficial effect [37]. Although this study showed that d-tagatose can promote biofilm formation of *S. lactis*, *L. rhamnosus GG* and *L. paracasei*, extracellular proteins, exopolysaccharides (EPS), and extracellular DNA (eDNA), which adhere to the outside of the cell, are key factors in bacterial colonization [38]. Therefore, additional studies of the above mechanisms are needed using multiomics analysis, including transcriptomics and metabolomics etc.. Based on above research, this study selected synbiotics composed of d-tagatose and *L. rhamnosus GG* to treat *Gallus gallus*' oxidative stress and inflammation conditions.

*Lactobacillus* able to improve the health of the intestine in animals [38], and has shown promise for treating intestinal flora disorders in animals [39]. The effects of synbiotics on the intestinal bacteria genera revealed that the abundance of *Phascolarctobacterium*, *Ligilactobacillus*, and *Bacteroides* was upregulated, and that of *Prevotellaceae* and *Desulfovibrio* was downregulated influenced by synbiotics consisting of d-tagatose and *L. rhamnosus GG*. As a reported that *Bacteroides* play an essential role in the processing of complex molecules to simpler such as promoting cellulose hydrolysis and digestion of nutrients [40], and they are closely related to biological processes, such as the biotransformation of minerals, bile salts, and steroids, therefore, this study believes that the increased abundance of *Bacteroides* is responsible for the enhanced weight gain, growth rate, and feed efficient rate of *Gallus gallus*. It is reported

that *Prevotellaceae* and *Desulfovibrio* can cause bacterial vaginosis [41], intestinal inflammation [42], suppurative arthritis [43] and ulcerative colitis [44], in this study, compared with the blank group, the abundance of *Prevotellaceae* and *Desulfovibrio* in the model group was significantly upregulated influenced by AFB1, and the abundance of *Prevotellaceae* and *Desulfovibrio* was downregulated by synbiotics treatment. Therefore, using synbiotics consisting of d-tagatose and *L. rhamnosus GG* can improve animal immunity, reduce disease incidence, and improve feed use efficiency by regulating cecal bacterial genera.

The cecal microbiota can produce a variety of metabolites by selectively fermenting exogenous undigested dietary components. Numerous beneficial metabolites in the cecum of *Gallus gallus* are upregulated, harmful metabolites are down-regulated influenced by synbiotics. The 2-arachidonoyl glycerol content in the cecal samples was increased with the increase of d-tagatose in the synbiotics, it can inhibit the proliferation of prostate cancer cell lines and pituitary tumor cells [45], as well as the apoptosis of hypothalamic neurons, but previous studies have shown that excessive amounts of 2-arachidonoyl glycerol disrupt the impairment of sperm motility [45], therefore, whether the upregulation of 2-arachidonoyl glycerol influenced by synbiotics is beneficial or harmful still requires in-depth research. Previous studies have shown that inosine is an anti-inflammatory drug that can alleviate fever and pain caused by lipopolysaccharide [46], In comparison with the model group, inosine content was up-regulated, and the mRNA expression level of *IL-1*, *IL-2*, *IL-4*, *IL-6*, *IgA*, *IgM* were down-regulated in treatment group, this indicates that inosine has may being involved in the inflammation treatment of *Gallus gallus*. Studies have shown that up-regulated biliverdin can ameliorate the oxidative stress response, inflammatory response, and apoptosis, it can also reduce free radical production in human umbilical vein endothelial cells (HUVECs) by the Nrf2/HO-1 signaling pathway mediated by PI3K/Akt [47], in this study, the gene expression level of HO-1 was significantly down-regulated after synbiotic treatment, this may be caused by d-tagatose promotes the biliverdin's synthesis by beneficial intestinal bacteria, thereby activating the Nrf2/HO-1 pathway in *Gallus gallus*. The up-regulation of 2-arachidonoyl glycerol, inosine, and biliverdin indicates that synbiotics can promote microbial activity to generate anti-inflammatory, antioxidant, and antineoplastic substances. Bilirubin is a marker of major depression [48], as well as retinopathy caused by diabetes mellitus type 2 (T2DM) [49]. In the present study, bilirubin content in the cecum was reduced gradually with the increase of d-tagatose content in the synbiotics. This finding indicates that synbiotics composed of d-tagatose and *L. rhamnosus GG* can potentially treat depression and retinal disease.

## Conclusion

RT-PCR, cecum bacteria genera analysis, and cecum metabolomics indicate that the synbiotics composed of d-tagatose and *L. rhamnosus GG* can treat inflammation and oxidative stress. Symbiotics can stimulate intestinal probiotics to produce beneficial metabolites by improving the abundance of the intestinal bacteria genera. Although the enhanced metabolic profiles are beneficial to animals, the association between metabolites and bacteria and the beneficial mechanism of metabolites requires further study. Overall, Synbiotics composed of d-tagatose and *L. rhamnosus GG* represent a promising approach to treating inflammation and oxidative stress. Nevertheless, the mechanisms underlying the treatment of oxidative stress and inflammatory responses remain to be further investigated and delved into.

## Supporting information

**S1 Table. Primer sequence used to detect oxidative and immune stress factors.**
(DOCX)

**S2 Table. Primer sequences used to detect the content of intestinal bacteria.**
(DOCX)

## Acknowledgments

This research was completed with the collaboration of all members of the Industrial Microbiology Laboratory. Here, this study would like to express our gratitude to Jie Chu, Xiaoxiao Zhang, Xuan Li, Aijiao Yin in the Industrial Microbiology Laboratory, Institute of Biology, Shandong Academy of Sciences.

## Author Contributions

**Conceptualization:** Yuanqiang Lv, Jie Chu.

**Funding acquisition:** Jie Chu.

**Investigation:** Xiaoxiao Zhang, Xuan Li, Aijiao Yin.

**Methodology:** Yuanqiang Lv, Jie Chu, Aijiao Yin.

**Project administration:** Jie Chu.

**Software:** Yuanqiang Lv.

**Supervision:** Jie Chu.

**Validation:** Xuan Li, Aijiao Yin.

**Visualization:** Yuanqiang Lv.

**Writing – original draft:** Yuanqiang Lv.

**Writing – review & editing:** Jie Chu.

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
