## [Decision Letter · Decision Letter 0]

13 Dec 2024

PONE-D-24-49947Synbiotics effects of d-tagatose and Lactobacillus rhamnosus GG on the inflammation and oxidative stress reaction of Gallus gallus based on the genus of cecal bacteria and their metabolitesPLOS ONE

Dear Dr. Chu,

Thank you for submitting your manuscript to PLOS ONE. After careful consideration, we feel that it has merit but does not fully meet PLOS ONE’s publication criteria as it currently stands. Therefore, we invite you to submit a revised version of the manuscript that addresses the points raised during the review process.

We look forward to receiving your revised manuscript.

Kind regards,

Mohammed Fouad El Basuini, Professor

Academic Editor

PLOS ONE

Journal Requirements:

2. Please include a caption for figure 3a.

3. Please include a caption for table 1.

[This study was supported by funding from the Key innovation Project of Qilu University of Technology (Shandong Academy of Sciences) [No.: 2024ZDZX03]. ]

 [The author(s) received no specific funding for this work.]

5. We note that you have indicated that there are restrictions to data sharing for this study. PLOS only allows data to be available upon request if there are legal or ethical restrictions on sharing data publicly. For more information on unacceptable data access restrictions, please see http://journals.plos.org/plosone/s/data-availability#loc-unacceptable-data-access-restrictions. 

Additional Editor Comments:

Line 22: The term "Abstracth" should be corrected to "Abstract."

Line 23: The manuscript mentions "L. paracasei" and "S. lactis" in the abstract, but the main focus seems to be on "L. rhamnosus GG." Clarify the role of these other strains.

Line 27: The phrase "The results indicated that oxidative and immune stress factor gene expressions quantity in Gallus gallus decreased significantly after 14 days of treatment" is vague. Specify which genes were measured and how their expression was quantified.

Line 35: The statement "Since China banned antibiotics in animal feed, the chicken mortality rate has increased from 2.8% to 4.2%" needs a citation for verification.

Line 49: The phrase "unabsorbed d-tagatose tilized by microbial colonizing in the colon" should be corrected to "unabsorbed d-tagatose is utilized by microbes colonizing the colon."

Line 56: The hypothesis "we hypothesize that tagatose’s in vivo probiotic properties remain to be elucidated" should be supported by a rationale or preliminary work.

Line 65: The source of the seven-day-old Gallus gallus should be described in more detail, including any specific breed or strain information.

Line 78: The experimental group is described as containing 2% d-tagatose, but the rationale for choosing this concentration is not provided. Include justification or references.

Line 107: The ethics statement should include more details about the specific guidelines followed for animal handling and care.

Line 116: The experimental design mentions "1 × 10^5 cfu/mL L. rhamnosus GG," but the concentration seems low for effective treatment. Provide justification or references for this dosage.

Line 123: The experimental design figure (Fig 1) should be referenced more clearly in the text, explaining its relevance to the study.

Line 130: The formula for weight gain rate (WGR) is missing proper formatting and should be corrected for clarity.

Line 173: The results section lacks statistical analysis details. Include information on the statistical tests used and their significance levels.

Line 189: The figure legend should specify the number of biological replicates used for each experiment. The axis labels in Fig 2a and 2b should include units (e.g., OD600, concentration).

In Fig 2c and 2d, the statistical significance between groups should be indicated with asterisks or other markers.

Line 191: The description of the weight gain rate (WGR) and specific growth rate (SGR) calculations should include the units used for each measurement.

Figure 3 (Line 192):

Line 192: The figure legend should explain the abbreviations used (e.g., WGR, SGR, FER). The units for weight gain rate (WGR) and specific growth rate (SGR) should be included in the axis labels (Fig. 3).

Line 214: The description of the bacterial genera analysis should include more details about the sequencing platform and methods used for data analysis.

Line 235: The metabolomics analysis section should specify the type of mass spectrometry used (e.g., LC-MS/MS) and the software used for data analysis.

Line 274: Figure 7 The volcano plots (Fig 7a and 7b) should include labels for the most significantly altered metabolites. The heat maps (Fig 7c and 7d) should include a color scale to indicate the range of metabolite concentrations.

Line 294: The discussion mentions "Kho et al. [16]" but does not provide a full reference. Ensure all references are complete and correctly formatted.

Line 349: The conclusion states that synbiotics can treat inflammation and oxidative stress, but this claim needs to be tempered with limitations and suggestions for further research.

Reviewers' comments:

Reviewer's Responses to Questions

**Comments to the Author**

1. Is the manuscript technically sound, and do the data support the conclusions?

Reviewer #1: Partly

Reviewer #2: Yes

2. Has the statistical analysis been performed appropriately and rigorously? 

Reviewer #1: I Don't Know

Reviewer #2: Yes

3. Have the authors made all data underlying the findings in their manuscript fully available?

Reviewer #1: Yes

Reviewer #2: Yes

4. Is the manuscript presented in an intelligible fashion and written in standard English?

Reviewer #1: No

Reviewer #2: Yes

5. Review Comments to the Author

Reviewer #1: Highlights should be written after abstract and before the introduction

Keywords should be written after the abstract not before

There are no recommendations ?

There are no authors contributions ?

There are no acknowledgement ?

There are no plan for the study area ?

The most descriptive methodologies are without any references--how ?

What about the clinical signs of the investigated birds due to Aflatoxin

Which type of Aflatoxin did you use ?

Aflatoxin is potent carcinogenic , at least induce necrosis not inflammation

Why you did not do a histological sections for more confirmation

What about the percentages of mortalities ?

Introduction should be more concise

Results should be rewritten

Discussion should be debate the obtained results with others

It is a very long paper

The IACUC-code should be written with M&M

Abstract :

There are no highlights

Abstract should contain --backgrounds/methods/results and conclusion

Abstract is very short

LN/24--what is /are the differences between probiotics/prebiotics and synbiotics (in table---role/mechanism --etc--with reference ) ???

LN/25--inflammatory conditions---mention all

LN/26--rewrite again (should write oxidative stress and immune response --etc )

LN/32--add poultry industry / intestinal health /probiotics to the keywords

Introduction

Introduction should be more concise

Aims need to be more clarified

Novelty of this study should be more highlighted and more adjusted

Materials and methods :

There are no plan for the study area

The most descriptive methods are without references --why ???

What about the LD50 of Aflatoxin-----?

LN/65--delete experimental

LN/66-70---what about systems of housing (watering/lighting--etc)

Aflatoxin is potent carcinogenic not inflammation induce, at least induce necrosis

LN/107--write IACUC code --should be

All experimental groups should be tabulated and done according to whom !!!

All item of great style writing --why ??

Results

All descriptions under figures should be more summarize

Should be rewritten again

What about the clinical signs of the investigated birds due to Aflatoxin

Aflatoxin is potent carcinogenic , at least induce necrosis not inflammation

Why you did not do a histological sections for more confirmation

What about the percentages of mortalities ?

The pathologic-scores lesions helps in comparison between groups

Discussion :

Should be based upon debating the obtained results with those of the previous investigators

References :

Some cited references need to be more updated

As volume/issue/pages/number--available----so no need for the link(s)---apply for all

Huge number of references were used

There are no gross figures ?

Reviewer #2: Comments to Author:

Minor points

1- In abstract, 21line: you wrote Abstracth, kindly convert it to Abstract.

2- In abstract, 23 line and line 27: you wrote (We)!!! The rule of manuscript

writing is to avoid using (We). So you should delete (We) and use formal

academic words (This study or The current study or The present study).

Kindly focus in this manuscript 14 times you wrote (We)!!! Convert all of

them to formal academic words (This study or The current study or The

present study).

3- In introduction, 33line: You wrote the abbreviation of the names of bacteria

such as L. lactis and L. plantarum; you should write the whole scientific

names of them (Lactococcus lactis; Lactobacillus plantarum) to be obvious

to the reader because here (in introduction) you used these names of bacteria

for the first time while in next sentences you can use the abbreviations only.

4- In materials and methods; you did not mention the period of sample

collection or the period of this study!!! Kindly mention the period.

5- In materials and methods; 82 and 83 lines: you wrote LB medium and

MRS plates you should write the whole names of the agars: Luria Bertani

(LB) agar and De Man–Rogosa–Sharpe (MRS) plates not only the

abbreviation to be obvious to the reader because here you used these names

of agars for the first time while in next sentences you can use the

abbreviations only.

6- In 90line; you wrote DPPH clearance, you should write the whole word

Diphenylpicrylhydrazine (DPPH) not only the abbreviation to be obvious

to the reader because here you used this word for the first time while in next

sentences you can use the abbreviation.

7- I suggest to add list of abbreviations in current manuscript.

Kind regards

6. PLOS authors have the option to publish the peer review history of their article (what does this mean?). If published, this will include your full peer review and any attached files.

Reviewer #1: **Yes: **Elsayed Eldeeb Mehana Hamouda

Reviewer #2: No

---

## [Author Response · Author response to Decision Letter 0]

27 Dec 2024

Replies to specific reviewers and editors can be found in the file "Response to Reviewers".

---

## [Decision Letter · Decision Letter 1]

7 Jan 2025

Synbiotics effects of d-tagatose and Lactobacillus rhamnosus GG on the inflammation and oxidative stress reaction of Gallus gallus based on the genus of cecal bacteria and their metabolites

PONE-D-24-49947R1

Dear Dr. Chu,

We’re pleased to inform you that your manuscript has been judged scientifically suitable for publication and will be formally accepted for publication once it meets all outstanding technical requirements.

Kind regards,

Mohammed Fouad El Basuini, Professor

Academic Editor

PLOS ONE

Additional Editor Comments (optional):

Reviewers' comments:

Reviewer's Responses to Questions

**Comments to the Author**

1. If the authors have adequately addressed your comments raised in a previous round of review and you feel that this manuscript is now acceptable for publication, you may indicate that here to bypass the “Comments to the Author” section, enter your conflict of interest statement in the “Confidential to Editor” section, and submit your "Accept" recommendation.

Reviewer #1: All comments have been addressed

Reviewer #2: All comments have been addressed

2. Is the manuscript technically sound, and do the data support the conclusions?

Reviewer #1: Yes

Reviewer #2: Yes

3. Has the statistical analysis been performed appropriately and rigorously? 

Reviewer #1: Yes

Reviewer #2: Yes

4. Have the authors made all data underlying the findings in their manuscript fully available?

Reviewer #1: Yes

Reviewer #2: Yes

5. Is the manuscript presented in an intelligible fashion and written in standard English?

Reviewer #1: Yes

Reviewer #2: Yes

6. Review Comments to the Author

Reviewer #1: The paper is fall in the scope and aims of PLOS One

As the authors did my all corrections

The paper accepted for publication

Reviewer #2: Greetings dear authors

Great job. Thanks alot for editings.

I hope all the best for you.

Kind regards.

7. PLOS authors have the option to publish the peer review history of their article (what does this mean?). If published, this will include your full peer review and any attached files.

Reviewer #1: **Yes: **Elsayed Eldeeb Mehana

Reviewer #2: No

---

## [Editor Report · Acceptance letter]

13 Jan 2025

PONE-D-24-49947R1 

PLOS ONE

Dear Dr. Chu, 

I'm pleased to inform you that your manuscript has been deemed suitable for publication in PLOS ONE. Congratulations! Your manuscript is now being handed over to our production team.

Kind regards, 

on behalf of

Prof. Mohammed Fouad El Basuini 

Academic Editor

PLOS ONE